# Strongyloidiasis in Children Outside the Tropics: Do We Need to Increase Awareness?

**DOI:** 10.3390/microorganisms9091905

**Published:** 2021-09-08

**Authors:** Elisabetta Venturini, Lara Fusani, Antonia Mantella, Leila Bianchi, Alberto Antonelli, Carlotta Montagnani, Elena Chiappini, Michele Spinicci, Alessandro Bartoloni, Gian Maria Rossolini, Lorenzo Zammarchi, Luisa Galli

**Affiliations:** 1Infectious Diseases Unit, Meyer Children’s University Hospital, 50139 Florence, Italy; elisabetta.venturini@meyer.it (E.V.); leila.bianchi@meyer.it (L.B.); carlotta.montagnani@meyer.it (C.M.); elena.chiappini@unifi.it (E.C.); 2Department of Health Sciences, University of Florence, 50121 Florence, Italy; lara.fusani@unifi.it; 3Department of Experimental and Clinical Medicine, University of Florence, 50121 Florence, Italy; antonia.mantella@unifi.it (A.M.); albertoanton88@gmail.com (A.A.); michele.spinicci@unifi.it (M.S.); alessandro.bartoloni@unifi.it (A.B.); gianmaria.rossolini@unifi.it (G.M.R.); lorenzo.zammarchi@unifi.it (L.Z.); 4Microbiology and Virology Unit, Careggi University Hospital, 50134 Florence, Italy; 5Referral Centre for Tropical Diseases of Tuscany, Infectious and Tropical Diseases Unit, Careggi University Hospital, 50134 Florence, Italy

**Keywords:** *Strongyloides stercoralis*, neglected tropical disease, children, international adoptions, migrants, screening, ivermectin, eosinophilia

## Abstract

Strongyloidiasis belongs to the group of neglected tropical diseases, due to diagnostic difficulties and the lack of systematic screening. Studies on strongyloidiasis prevalence are often heterogenous and mainly performed in adults in endemic countries. We retrospectively enrolled 2633 children referred to a tertiary care hospital in Italy between 2009 and 2020 and tested for *S. stercoralis* infection. Sixty-one (2.3%) had a positive serology and for 55 of them, clinical and epidemiological information were available. Thirteen cases (24%) were diagnosed in Italian children without history residency or travel to foreign countries, while the remaining were internationally adopted or migrant children. Seropositive patients were mostly asymptomatic, and often eosinophilia was the only sign of strongyloidiasis. Sero-reactivity to *Toxocara canis* was found in 1/3 of patients. Ivermectin was used in 37 (75.5%) treated patients. A significant reduction of eosinophil levels and IgG titer was seen after treatment. Our study confirms that strongyloidiasis is usually asymptomatic in children. However, due to the ability of the parasite to cause a life-long infection together with the risk of a severe form in case of immunosuppression, it is important to identify and treat infected children. Special consideration should be reserved to high-risk groups, such as immigrants and international adoptees, where screening for *S. stercoralis* is indicated. However, the study highlights that sporadic cases of autochthonous strongyloidiasis in Italy may occur. Therefore, pediatricians should be aware of this condition, which is often under-recognized.

## 1. Introduction

Neglected tropical diseases (NTDs) are a group of various infections caused by bacteria, viruses, protozoa, and helminths, widespread in low- and middle-income countries, such as Africa, Asia, and South America [1]. In recent years, the increasing of migratory flow, adoption, and international travel has led to a rising relevance of NTDs also in developed countries [2]. In spite of this, NTDs are still often unrecognized, leading both to a delay in proper treatment and to an underestimate in their prevalence [3]. Globally, among the NTDs, strongyloidiasis caused by the soil-transmitted helminth *Strongyloides stercoralis* is one of the most commonly underdiagnosed, although a recent study estimated that more than 600 million people contract this infection globally [4].

At present, only few studies, mainly conducted in adults, have evaluated the prevalence of strongyloidiasis in immigrants, with conflicting results [5,6]. Literature is scarce about strongyloidiasis in childhood. Moreover, the majority of available studies have been conducted in endemic areas, but little is known about *S. stercoralis* prevalence in European and Italian children [7,8,9,10]. This is mainly due to the fact that a uniform and systematic screening for strongyloidiasis in children coming from high-burden countries, or in children with unexplained eosinophilia, is not usually performed.

This is a serious gap, considering that an increase of immigrants from Sub-Saharan Africa (SSA) has been reported in Italy, in addition to the foreign-born individuals already resident in our country (about 8% of the Italian population) [3,11]. Moreover, in the last 5 years, about 1 million children have migrated to Europe, without considering international adoption and children born of foreign parents [12].

In addition to the lack of a systematic screening for high-risk subjects, diagnostic difficulties are also present. In fact, coprological methods are not always able to identify the larvae, considering the intermittent fecal release of *S. stercoralis* [13,14]. Among the available tests, serology shows the highest detection rate in immunocompetent individuals, but it is poorly specific due to the possible cross-reaction with other parasites, being scarcely reliable in high-burden countries [15]. However, it is an excellent screening tool in low-endemic countries [15,16]. Regarding clinical signs and symptoms, strongyloidiasis can mimic other parasitic diseases. In fact, unspecific gastrointestinal, dermatological, and pulmonary manifestations are commonly reported, while failure to thrive could also be present in children [17,18]. Frequently, *S. stercoralis* infection can proceed asymptomatically, with peripheral eosinophilia being the sole sign [19]. Despite this, it is crucial to detect and treat both symptomatic and asymptomatic subjects, in order to prevent hyperinfection syndrome, which has a high mortality of around 87% [14,20]. Fortunately, strongyloidiasis can be easily treated with ivermectin, with high effectiveness and good tolerability [21]. A single dose of ivermectin is effective, thus increasing treatment adherence [22,23,24]. Regarding the pediatric age, ivermectin is approved in children over 15 kg, although a recent review highlights its high safety profile even below this weight [25]. Albendazole, which has been considered an alternative drug for treatment of strongyloidiasis, demonstrated unsatisfactory efficacy compared to ivermectin in several clinical trials [26,27,28,29].

Considering all these issues, the primary objective of our study was to describe the prevalence of *S. stercoralis* in a large cohort of children evaluated in the Infectious Diseases Unit of a tertiary care children’s hospital in Italy and screened for *S. stercoralis*. The secondary objective was to describe the demographic, clinical, and hematochemical characteristics of children with *S. stercoralis* infection and to evaluate the therapy administered and the therapeutic response.

## 2. Materials and Methods

### 2.1. Study Design

We retrospectively enrolled 2633 patients below 18 years of age, referred to the Infectious Diseases Unit of Anna Meyer tertiary care children’s University Hospital in Florence, Italy, between 1 January 2009 and 31 December 2020, and tested for *S. stercoralis* infection. Within our Infectious Disease Unit, *S. stercoralis* serology is usually performed in case of clinical suspicion or as part of migration health assessment. According to the local ethical board, all parents had signed, at the time of their first hospital access, an informed consent for children’s data inclusion in observational studies with anonymized data extraction.

For each patient, data about age, gender, and country of birth were collected.

Children were then divided in three groups depending on *S. stercoralis* serologic results: negative, borderline, or positive (Figure 1).

For positive patients, clinical signs and symptoms, as well as past medical history, were evaluated. Blood and fecal samples were obtained to perform serologic assays for other NTDs (in accordance with patients’ risk factors) and parasitological examination or stool culture, to investigate the possible co-infection with other parasites. Moreover, eosinophil cell count and IgE level, before and after treatment, were determined. Treatment information and response to target therapy were also assessed.

### 2.2. Definitions

Strongyloidiasis was diagnosed in children with positive serologic results or stool test for *S. stercoralis*, regardless of eosinophil levels. After starting proper therapy, the infection was judged as “cured” in patients whose *S. stercoralis* serologic test turned negative during follow-up or, alternatively, in case of a significant decrease of eosinophil levels (halved after therapy compared to baseline values) and of *S. stercoralis* serology index (IgG ratio after/before treatment < 0.6) [30,31,32]. The criteria for “therapeutic failure” were if the worm was identified in stool samples or if the patient has a persistently positive serologic result 12 months after treatment. We defined possibly cured” as children with a negative coprological test and normalized eosinophil levels, without serological follow-up after treatment.

Peripheral eosinophilia was defined by an eosinophil cell count above 500 cells/µL. Therefore, it was classified as mild (500–1500 cells/µL), moderate (1500–5000 cells/µL), or severe (>5000 cells/µL), according to eosinophil levels [33].

### 2.3. Laboratory Methods

Microbiological analyses were performed at the Microbiology and Virology Laboratory at Careggi University Hospital in Florence, Italy. Microscopic stool examination and/or agar plate stool culture were performed as previously described [34].

Several serological enzyme-linked immunosorbent assay (ELISA) commercial kits have been used over the years. The first ELISA kit, used until September 2016, was produced by DRG Instruments. In accordance with the manufacturer’s instructions, the index (optical density (OD)/relative light units (RLU)) was not reported, but the result was expressed as positive or negative. Then, the *Strongyloides ratti* ELISA kit from Effegiemme S.R.L (product by Bordier Affinity Products SA, Crissier, Switzerland) was performed. IgG index ≤ 0.9 was defined negative, 0.9–1.1 borderline, and ≥1.1 positive.

### 2.4. Statistical Analysis

Data were analyzed by SPSS. Statistics 24.0 statistical software. Categorical variables and frequencies were compared by means of the χ^2^ test or Fisher test, as appropriate. Quantitative variables were reported as median and interquartile range (IQR) and compared by means of nonparametric tests (Mann–Whitney U). A P value of 0.05 was considered to indicate statistical significance.

## 3. Results

Overall, 2633 children evaluated at the Infectious Disease unit during the study period were screened for *S. stercoralis* infection. Overall, 1543 out of 2633 participants were male (58.6%) and the mean age was 7 years (IQR 4.19–10.18). Most of them (2288/2633, 86.7%) were referred to our center to perform screening for recent immigration or international adoption.

Enrolled children were mainly from Oriental Europe and Balkan peninsula (682/2633, 25.9%), followed by Africa (472/2633, 18%), Latin America (493/2633, 18.7%), and Asia (443/2633, 16.8%). Only 345/2633 (13.1%) children were born in Italy, and among them 125/345 (36.2%) had foreign parents. Forty-six (46/2633, 1.7%) children came from other European countries. For 152/2633 (5.8%), mainly adopted children, we could not define the country of birth. The countries of origin of all participants are summarized in Figure 2.

Overall, 96.6% (2544/2633) children had negative serological results at the first evaluation. Thirty participants had a borderline index (30/2633, 1.14%), so underwent a second test and then were followed up according to the result. Overall, 59/2633 (2.24%) children had a positive serology at the first test, while two patients turned out to be positive after an initially borderline result, with an overall prevalence of 2.3% in our cohort.

### 3.1. Assessment of Children with Borderline Serology

Among the children with borderline serology, 6 were lost at follow-up (6/30, 20%), and 24/30 were retested. Of those, 19 (19/24, 79.1%) showed a negative index for *S. stercoralis*; two (2/24, 8.3%) turned out to be positive and then were treated accordingly, while 3 (3/24, 12.5%) confirmed a borderline result. Almost 76.7% (23/30) were immigrants or adoptees, screened for *S. stercoralis* as part of migration health assessment. Three Italian children were screened for strongyloidiasis due to a mild eosinophilia. For the other four children (4/30, 13.3%), the reason for being tested was unknown. All patients, except for one of them, were tested for *S. stercoralis* on stool samples and turned out to be negative.

Age was significantly higher in children with borderline serology compared to positive and negative ones (10 years IQR: 6.7–11.5 vs. 7.1 years IQR: 4.49–10.59 vs. 6.9 IQR: 4.16–10.15 respectively; *p* = 0.014).

Peripheral eosinophil counts were within the normal range, except for three patients: two children who presented with eosinophilia (one infected by *Schistosoma haematobium* and another with elevated IgE), and one had moderate eosinophil counts but also had a positive serology for *Toxocara canis*.

Considering those with two consequently borderline index, the only one receiving treatment was a girl born in Chile who received two doses of ivermectin, though neither eosinophilia nor any symptoms were reported, with a subsequent negative index more than 12 months after treatment.

The clinical features and comorbidities of this group are represented in Table 1.

### 3.2. Assessment of Children with Positive Index

Overall, 61 children (61/2633, 2.3%) had a positive serology for strongyloidiasis.

In 78.6% (48/61) of cases, infection was detected due to screening programs for immigrants and adopted children. However, the positivity rate was significantly higher in children born in Italy by foreign parents (4%, 5/125), if compared to children born in Italy from Italian parents (3.6%, 8/220), adoptees (2.5%, 43/1709), and immigrants (0.86%, 5/579) (*p* < 0.001).

Among them, 37/61 (60.6%) were male. The majority of positive patients came from Africa (18/61, 29.5%), followed by Asia (14/61, 22.9%), Italy (13/61, 21.3%), Latin America (11/61, 18%), and Oriental Europe (5/61, 8.2%) (*p* = 0.013).

The complete clinical data were available only for 55 patients. Only 3/55 children (5.4%), all immigrants, reported overseas travel to their country of origin after their first resettlement in Italy. Of note, the 13 seropositive patients born in Italy probably acquired the infection in Italy since no history of travel to foreign countries was clearly referred to during the clinical evaluation. Among them, 2/13 (15.4%), born by foreign parents, had a positive serology because of the transplacental passage of maternal *S. stercoralis* IgG. They did not receive any therapy and were followed up until serology negativization.

Clinical records were not available for 3/13 (23%) children. Data about these cases are reported in Table 2.

A minority of patients analyzed showed signs and/or symptoms consistent with *S. stercoralis* infection (Table 3).

Nine patients (9/55, 16.4%) also had positive stool tests for other parasites that could mimic symptoms of strongyloidiasis and 17/55 (30.9%) showed a positive serology for *Toxocara canis*. Of them, 10/17 (58.8%) were tested through Western blot analysis.

Almost half of them (25/55, 45.5%) had other comorbidities (Figure 3).

Most of the patients analyzed (37/55, 67.3%) presented, at the first evaluation, with peripheral eosinophilia (median 750 cells/µL, IQR: 301–1827). Specifically, in 59.4% of them (22/37), mild eosinophilia was registered; in 37.8% (14/37), a moderate one; and in only 1/37 (2.7%), severe eosinophil levels. Among them, 11/37 (29.7%) also had a positive serology for *Toxocara canis* and, of these, 6/11 (54.5%) showed a mild eosinophilia and the others (5/11, 45.5%) a moderate one.

IgE levels were also evaluated in 40/55 patients, showing a median value of 317.50 (IQR: 68.25–786.75) UI/L and being within the normal range for age in 65% (26/40) of the cases. Microscopic parasitological examination of the stools was available for all the children with positive serology for *S. stercoralis*. Overall, 9/55 (16.4%) children were positive for other parasites, mainly for *Giardia intestinalis* (4/9, 44.4%) and *Hymenolepis nana* (3/9, 33.3%).

Two patients (2/55, 3.6%) were positive for *S. stercoralis* to microscopic examination. Stool culture for *S. stercoralis,* before starting therapy, was requested in 19 patients, with positivity in only in 3/19 patients (15.8%). All tested negative to the microscopic examination. Among those children with a *S. stercoralis*-positive stool test, 1/5 (20%) was also infected by *Giardia intestinalis*, while another child (1/5, 20%) had a polyparasitism. All of them had an increased eosinophil count, 2/5 (40%) showed a mild eosinophilia, and the remnant (3/5, 60%) a moderate one. As part of the screening program for adopted or immigrant children, or according to the clinical history of abdominal symptoms, all positive patients were also subjected to an abdominal ultrasound and 38/55 (69.1%) to a chest X-ray without finding any abnormalities.

### 3.3. Assessment of Therapeutic Response of Patients with Positive Serology for S. stercoralis

Information on the administered therapy was collected. Overall, 13/55 (23.6%) children received other antiparasitic drugs prior to the initiation of strongyloidiasis treatment, mainly tinidazole (5/13, 38.5%) for giardiasis. Six patients (6/55, 10.9%) did not receive target therapy for strongyloidiasis. In particular, 33.3% of them (2/6) were lost at follow-up, and 1/6 (16.7%) were admitted to an adult hospital for the continuation of care, as he reached the age of the majority. One girl (1/6; 16.7%) with hepato-splenic calcifications was not treated, but, at the clinical evaluation more than 12 months later, a negative serology was found.

Among patients who received treatment (49/55), 37/49 (75.5%) received ivermectin. All these children were over 15 kg. The dosage per dose was 200 mcg/kg, which was prescribed for two consecutive days in 25/37 patients (67.6%). Two doses of ivermectin 15 days apart and 4 doses of ivermectin (administered on days 1, 2, 15, and 16) were prescribed in 6/37 (16.2%) and 4/37 (10.8%) children, respectively. A single dose was administered only in 2/37 children (5.4%). One girl treated with ivermectin (1/37, 2.7%) reported 48 h after the administration of the second dose of ivermectin drowsiness, fatigue, inappetence, and slight fever. No other adverse events were reported.

Twelve patients (24.5%) received albendazole 400 mg twice a day for 7 days. More than half of them (8/12) were seropositive for *Toxocara canis*. Only one girl, aged 8 months, was treated with albendazole because ivermectin was contraindicated.

After taking the appropriate therapy, eosinophil levels showed a decreasing trend, as shown in Figure 4. Unfortunately, not all enrolled patients performed blood tests at the same time. Interestingly, the eosinophil count was significantly lower at 6 months since treatment started (*p* = 0.032), whereas no differences were detected at 3 and 12 months (*p* = 0.842, *p* = 0.22, respectively).

Regarding serologic negativization, for 19/49 patients treated (38.7%), we had no available data and 4/49 (8.16%) are still in follow-up. Twenty-six patients (26/49, 53%) had a negative serologic test after therapy, of them 17/26 (65.4%) within 6 months of treatment and the remnant (8/26, 30.8%) after 6 months. The *S. stercoralis* IgG index was available for 54/61 patients at diagnosis (88.5%) with mean values (1.77, IQR: 1.36–3.02). Interestingly, a statistically significant reduction was registered during follow-up at 3, 6, 12, and more than 12 months after therapy (0.8, IQR: 0.29–1.42; 0.58, IQR: 0.24–1.38; 0.96, IQR 0.28–1.42; 0.45, IQR 0.28–0.72; respectively, *p* < 0.001) (Figure 5).

A negative antibody index was reached faster in children treated with ivermectin than with albendazole, though no statistically significant differences were observed (*p* = 0.08). Similarly, no significant differences were reported in the distribution of eosinophils between those who had a negative serologic test before or after 6 months of therapy (*p* = 0.93).

Treatment with ivermectin was repeated in 8/49 patients (16.3%) due to a positive titer at 3 (2/8, 25%), 6 (3/8, 37.5%), and 12 months (2/8, 25%) after the first drug administration. Two of them (2/8, 25%) became negative for strongyloidiasis whereas the other 6/8 were lost to follow-up without retesting the serologic test. Among them, 2/8 (25%) were also tested for HTLV infection, and both turned out to be negative. HIV serology was required only as part of screening for adoptees and immigrants, and none of the enrolled patients were positive.

## 4. Discussion

Herein, we described a cohort of 2633 patients, mainly adopted or immigrated children, that were seen in a tertiary care children’s hospital in Italy, and were screened for strongyloidiasis by serologic testing. In this population, the *S. stercoralis* serologic test was positive in 2.3% of the children. To our knowledge, this retrospective cohort study is the first study evaluating the positivity rate of *S. stercoralis* in a group of children in Italy.

Our population included mainly boys (60.6%), with a mean age of 7 years. No correlation was found between strongyloidiasis and age or sex in our cohort, even if a recent study on Malaysian children reported a significative relation with male gender and older age [35]. In our study, the African origin was significantly correlated with *S. stercoralis* infection, compared to other countries (*p* = 0.013). Interestingly, we found 13 cases of seropositive children, without recent history of travel outside Italy, considered as autochthonous transmission. Autochthonous cases in Italy and in the Mediterranean basin are usually observed in elderly who acquired the infection in the past when fecal contamination of the environment was common. However, sporadic cases are still observed, and several hypotheses can explain these recent cases, including soil-transmitted cases sustained by a persistent free-living cycle in the environment; canine or, less likely, human reservoir; or direct interhuman transmission (for example through sexual intercourse) [36].

Moreover, a higher positivity rate in children born in Italy, both in those with foreign parents and in Italian children, compared to adoptees and immigrants (4% and 3.6% vs. 2.5% and 0.86%, respectively) was found. This difference could be due to the fact that, in Italian children, *S. stercoralis* was searched in case of clinical suspicious, such as unexplained high eosinophil levels. On the other hand, adoptees and immigrants were screened for strongyloidiasis regardless of the presence of clinical signs. This could lead to an overestimation of the prevalence in Italian children in our study. The fact that children born in Italy from foreign-born parents showed a seroprevalence higher than autochthonous children with Italian parents cannot exclude intra-familial transmission. However, it seems unlikely, since direct human-to-human transmission has been observed only through sexual exposure or organ transplant [37]. Furthermore, it is possible that children travelled to their parents’ home country, but no specific data were available in their clinical records.

Only few studies, mainly performed on adults, assessed the impact of strongyloidiasis in Europe, showing a higher rate of positivity [5,38]. In particular, Martelli et al. conducted a multicenter cross-sectional study and assessed in a group of 930 immigrants, a strongyloidiasis seroprevalence around 4.5% [5]. In Spain, a recent observational study on immigrants and international travelers reported a positivity rate for *S. stercoralis* infection of up to 9.7% [38]. A similar rate of positivity was found in a recent meta-analysis on strongyloidiasis prevalence among migrants (both children and adults) who live in non-endemic countries, with a seroprevalence of 12.2% (95% CI 9.0–15.9%; I^2^ 96%) and a prevalence of positive stool samples of around 1.8% (1.2–2.6%; 98%) [6]. The higher rate of serologic positivity if compared to our study could be related to the higher chance of adults being infected by *S. stercoralis*.

Regarding the pediatric age, limited studies are currently available only in endemic regions, showing a significantly higher seroprevalence compared to our study. In particular, in Angola, Cambodia, and Malaysia, the strongyloidiasis positivity rate of children was 21.4%, 24%, and 15.8%, respectively [7,35,39].

This discrepancy of the infection rate between endemic and non-endemic regions could be due not only to a different burden of the disease, but also to test limitations. In fact, a well-known limit of the serology for *S. stercoralis* is the high rate of false positives due to cross-reaction with other parasites that can increase the positivity rate in low-income countries where parasitosis are widespread [7]. On this matter, Bisoffi et al. underlined that to define a patient as positive, a higher serologic cut-off could be helpful, increasing specificity and guaranteeing a good sensitivity at the same time. In particular, for ELISA tests, they suggested setting the limits to ≥2.5 and ≥2.2 for IVD and Bordier kit, respectively [40]. A similar cut-off has been proposed in a recent study conducted on immunosuppressed adult patients [41]. However, these cut-offs have not been validated in clinical practice, and further studies are certainly demanded. Moreover, in our population, the agar plate stool test was performed only in a sub-group of patients with positive serologic assay. Therefore, our prevalence could be underestimated, especially if compared with a recent study conducted in Spain that combined serology with fecal tests [38].

In the sub-group of children with positive serology, the agar plate stool test confirmed its importance for the confirmation of active infection. The low positivity rate (15.8%) found in our study could be explained by the use of only one coprological diagnostic technique, while Khieu V. et al., in Cambodia [7], combined three different methods on three fecal samples, reaching a positivity rate of around 24.5%.

In our study, full clinical data were available only for children with positive serology. The majority of them were asymptomatic at clinical evaluation, whereas the most common symptoms were pruritis or cough and wheezing (both 18.2%), according to literature data [8]. Unfortunately, these manifestations are nonspecific, also being found in other diseases. This makes it harder to correlate those symptoms with strongyloidiasis, in particular, when other underlying conditions, such as asthma, scabies, and other parasitic infections that could mimic strongyloidiasis, are reported.

High eosinophil levels are often the only sign of strongyloidiasis [19] and were detected in 67.3% of positive children at their first evaluation. The monitoring of peripheral eosinophilia was helpful also in the evaluation of therapeutic response. In fact, a significant reduction of eosinophils count was demonstrated 6 months after treatment. The lack of statistical significance of eosinophils decrease at 12 months could be explained by the smaller sample size. In our study, we considered eosinophils decrease as an indirect marker of *S. stercoralis* load reduction and, when combining this data with a negative coprological test, patients were considered as “probable cured” without repeating serology. The decrease of eosinophils has been considered a marker of therapeutic response also in previous studies [31,42]. In another study by Repetto at al., a significant decrease of eosinophils was reported a month after ivermectin treatment [42]. Moreover, Nuesch et al. found, in a group of 31 adults with strongyloidiasis, a statistically significant reduction of eosinophil levels three months after being treated. In this study, a significant reduction of eosinophils and *S. stercoralis* antibody index was considered as indicative of a successful treatment [31].

Regarding co-infections, positive serology for *Toxocara canis* was detected in around 30.9% of cases, whereas *Giardia intestinalis* and *Hymenolepis nana* in 44.4% and 33.3% of cases, respectively. In fact, it is well known that there is a high risk of polyparasitism in these patients [10,43]. Moreover, *Toxocara canis* serology could be influenced by other parasitic infections [44]. However, most patients with positive *Toxocara canis* serology were tested through Western blot techniques, in order to reduce the risk of cross-reaction with *S. stercoralis.*

Ivermectin was the first treatment choice in 67.3% of patients. A recent Cochrane published in 2016 [24] supports the use of ivermectin as the drug of choice in strongyloidiasis with a higher cure rate and similar safety profile compared with albendazole.

In our study, the most frequently used ivermectin regimen was one dose on two consecutive days (67.6%). Recently, a trial on ivermectin dosing in children was published, showing that one dose should be preferred over the four-dose regimen [22]. In our population, minor reversible side effects were reported only in one patient.

Ivermectin is not widely easy to obtain. In particular, in Italy, this drug is not commercially available and is usually obtained from other countries, causing treatment delays [45]. Moreover, Ivermectin is off label in children less than 15 kg, despite a systematic review published in 2021 that shows its safety in those patients [25]. In our study, patients with a weight below 15 kg were treated with albendazole, which is the second treatment choice.

A serologic assay was repeated after treatment in about half of the cases until negativization. In particular, negative titer was reached within 6 months in 65.4% of the cases. A statistically significant reduction of the *S. stercoralis* IgG index was detected at 3, 6, 12, and >12 months after treatment.

Children treated with albendazole (21.8%) had a slower negativization of the antibody index, and almost half of them required a second course with ivermectin due to a persistently positive titer.

We defined treatment failure in cases of a persistently positive index and/or evidence of *S. stercoralis* larvae on fecal samples despite administration of proper therapy. At the moment, there are no studies in the literature on the management of therapeutic failure and the need for a second treatment [38]. Further studies are needed in order to better define the timing for the follow-up and for the need for a second drug administration.

Our study has some limitations. First of all, it was retrospective. Due to this, full clinical and hematochemical data were available only for children with a positive serologic test. Therefore, it was not possible to establish which risk factors were associated with a higher rate of positivity, except for epidemiological data.

Similarly, the timing of blood tests was heterogeneous, leading to difficulties in making comparisons. Moreover, the fecal examination was performed only in patients with positive serologic test, and with different techniques over the years, with a subsequent possible underestimation of its positivity rate. Therefore, in our population, few stool samples (one to three for each patient) were collected. Some studies reported that a single stool exam is diagnostic in only one-third of patients, while serial stool examinations increase the sensitivity but may be impractical [46]. On this matter, Nielsen et al. estimated that seven stool exams would provide close to 100% sensitivity [47] Finally, the serological tests used, all based on the ELISA method, varied during the study period. Therefore, the antibody index has been available only since September 2016. Moreover, we are aware that our study was based mainly on serologic tests, which may have both false-positive and false-negative results; however, the fact that most patients presented a decline in the titer of serology after treatment suggests that they could have been infected.

In conclusion, our study confirms that strongyloidiasis in children is a benign condition in the large majority of cases. However, due to the risk of severe forms in the case of immunosuppression, it is important to identify and treat infected children. A special consideration should be reserved for high-risk groups, such as immigrants and international adoptees, where screening for *S. stercoralis* is indicated. Interestingly, the highest risk group included children born in Italy by immigrant parents, probably due to travel to country of origin. The higher risk of imported disease in visiting friends and families is also well described for other infectious diseases [48]. However, our study also underlines a significant risk in Italian children presenting with signs or symptoms of infection.

Therefore, pediatricians should be aware of this condition, which is often underrecognized. *S. stercoralis* serologic testing was confirmed to be a useful tool, especially in the initial evaluation in patients coming from endemic areas. A combination of tests (serology, stool culture, eosinophils count) is indicated in order to increase the diagnostic yield.

Larger prospective studies are needed in children in order to clearly define some aspects of strongyloidiasis, such as the management of treatment failure and long-term follow-up.

## Figures and Tables

**Figure 1 microorganisms-09-01905-f001:**
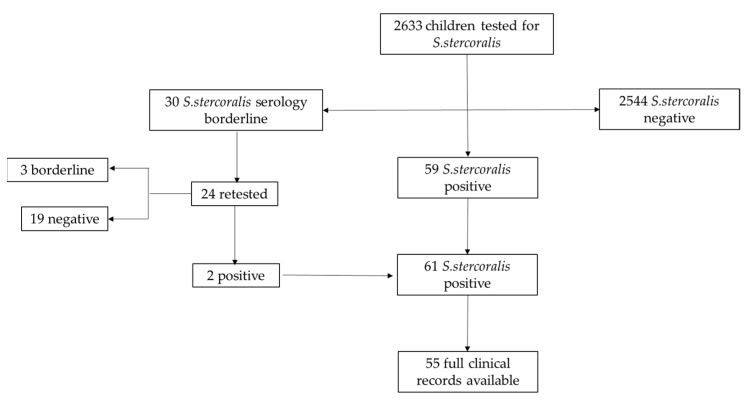
Flow chart of patients’ selection.

**Figure 2 microorganisms-09-01905-f002:**
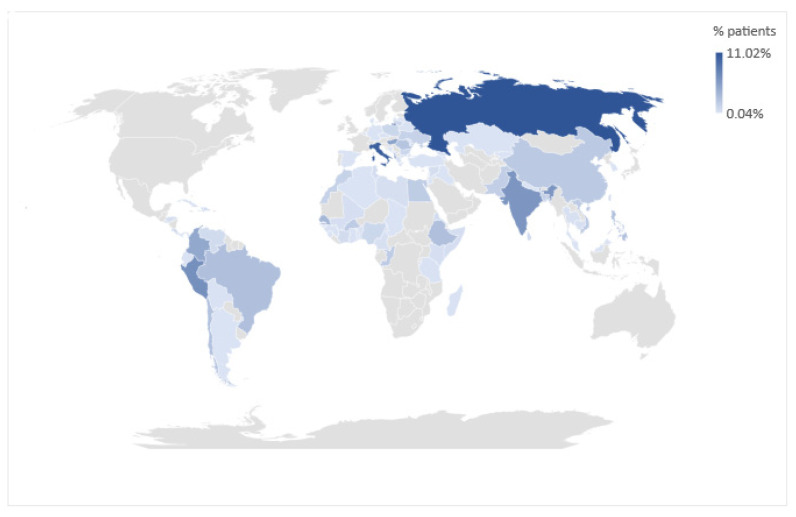
Geographic representation of participants’ country of origin.

**Figure 3 microorganisms-09-01905-f003:**
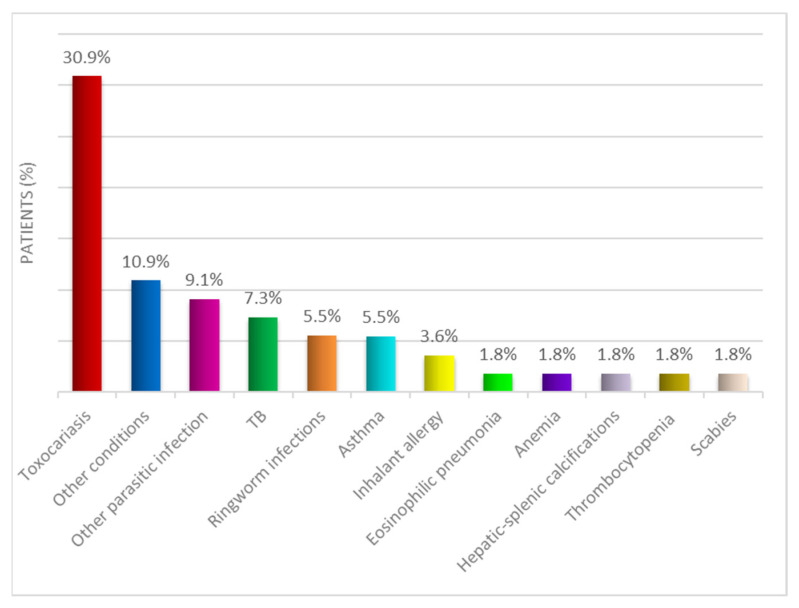
Comorbidities of patients with positive serology for *S. stercoralis*.

**Figure 4 microorganisms-09-01905-f004:**
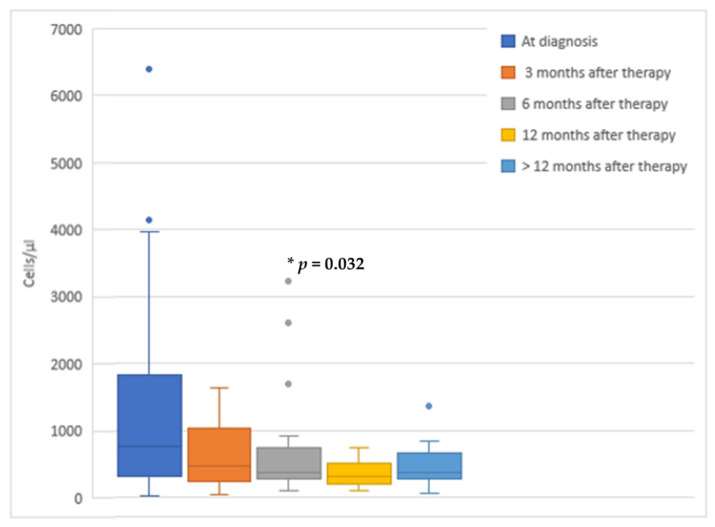
Eosinophil levels after the onset of proper therapy.

**Figure 5 microorganisms-09-01905-f005:**
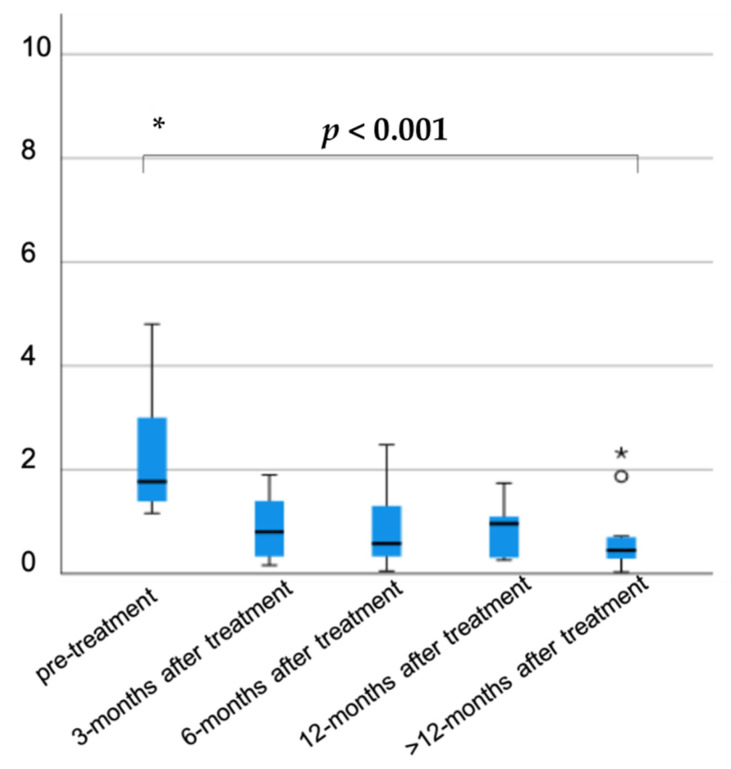
*S. stercoralis* IgG index during follow-up, according to treatment.

**Table 1 microorganisms-09-01905-t001:** Clinical features of patients with borderline *S. stercoralis* index.

Patient (Sex, Age)	Home Country	Eosinophil Level (Cells/µL)	Clinical Signs/Symptoms	Comorbidities	Result of *S. stercoralis* on Stool Samples	Result of *S. stercoralis* Serology (Second Retesting)
Patient 1 (M/8 years)	Colombia	225	No	No	Negative (ME)	Negative
Patient 2 (M/10 years)	India	456	No	Latent TB	Negative (ME)	Negative
Patient 3 (M/2 years)	Mali	153	No	Anemia	Negative (ME and AC)	Lost at FU
Patient 4 (F/11 years)	Colombia	1369	No	No	Negative (ME)	Positive
Patient 5 (M/3 years)	Thailand	1667	No	Toxocariasis	Negative (ME and AC)	Lost at FU
Patient 6 (F/3 months)	Italy	950	No	No	Negative (ME)	Negative
Patient 7 (M/10 years)	India	509	No	Latent TB	Negative (ME)	Negative
Patient 8 (M/11 years)	Italy	1480	No	No	Not performed	Negative
Patient 9 (M/12 years)	Pakistan	600	No	No	Negative (ME and AC)	Lost at FU
Patient 10 (M/11 years)	Italy	890	No	No	Negative (ME)	Negative
Patient 11 (M/6 years)	Pakistan	5000	No	elevated IgE value	Negative (ME)	Negative
Patient 12 (M/14 years)	Russia	360	No	No	Negative (ME)	Negative
Patient 13 (F/5 months)	Pakistan	690	No	No	Negative (ME)	Lost at FU
Patient 14 (F/6 years)	India	227	No	No	Negative (ME and AC)	Positive
Patient 15 (F/10 years)	Colombia	348	No	No	Negative (ME and AC)	Negative
Patient 16 (M/10 years)	Colombia	258	No	No	Negative (ME and AC)	Negative
Patient 17 (F/10 years)	Hungary	209	No	Giardiasis	Negative (ME and AC)	Negative
Patient 18 (F/9 years)	Kosovo	300	No	No	Negative (ME and AC)	Borderline
Patient 19 (F/10 years)	Chile	64	No	Giardiasis	Negative (ME)	Borderline
Patient 20 (F/17 years)	Romania	100	No	No	Negative (ME)	Negative
Patient 21 (M/11 years)	Senegal	220	No	Latent TB	Negative (ME)	Negative
Patient 22 (M/8 years)	India	189	No	No	Negative (ME)	Negative
Patient 23 (M/11 years)	Senegal	6575	Abdominal pain	*S. haematobium* infection	Negative (ME)	Borderline
Patient 24 (F/12 years)	Philippines	440	No	Latent TB	Negative (ME)	Negative
Patient 25 (F/16 years)	Nigeria	25	No	Latent TB	Negative (ME)	Lost at FU
Patient 26 (M/7 years)	Hungary	95	No	No	Negative (ME)	Negative
Patient 27 (F/15 years)	Romania	478	No	Toxocariasis	Negative (ME)	Negative
Patient 28 (M/11 years)	Peru	176	No	No	Negative (ME and AC)	Negative
Patient 29 (M/11 years)	Peru	510	No	Pleural TB	Negative (ME)	Negative
Patient 30 (F/6 years)	Italy	280	Not known	Not known	Negative (ME)	Negative

FU: follow-up; TB: tuberculosis, ME: microscopic examination, AC: agar plate culture.

**Table 2 microorganisms-09-01905-t002:** Clinical characteristics of Italian-born children without history of travel abroad with positive *S. stercoralis* serology.

Patient (Sex, Age)	Year of Diagnosis	Parents Home Country	Serology Titer	Eosinophil Count (Cells/µL)	Symptoms	*Toxocara canis* Serology	Other Co-Infections	Other Anti-Parasite Treatment	*S. stercoralis* on Stool	Therapy	Serology Negativization	Normalization of Eosinophil Count after Therapy
Patient 1 (F/6 years)	2011	Italy	3.1	585	pruritis, nonspecific skin eruption	Negative	Pinworms	Pyrantel pamoate	No (ME) ^1^	Ivermectin	Yes	Yes
Patient 2 (F/8 months)	2014	Italy	1.6	10,673	Vomiting, anorexia	Negative	No	No	No (ME)	Albendazole	Yes	Yes
Patient 3 (M/11 years)	2015	Italy	1.97	1128	Cough and wheezing	Negative	No	No	No (ME)	Ivermectin	Yes	Yes
Patient 4 (F/15 years)	2015	Italy	1.54	154	hepato-splenic calcifications	Negative	No	No	No (ME)	No	Yes	
Patient 5 (M/10 years)	2015	China	2	278	pruritis, urticaria	Positive	No	No	No (ME)	No	Unknown	
Patient 6 (M,9 years)	2016	Kosovo	1.63	34	Cough	Negative	No	No	No (ME and AC) ^2^	Ivermectin	Yes	Yes
Patient 7 (M,14 years)	2017	Italy	1.81	3959	Eosinophilic pneumonia	Negative	No	No	No (ME and AC)	Ivermectin	Yes	Yes
Patient 8 (F/6 years)	2019	Perú	1.46	1963	pruritis, abdominal pain	No	No	No	No (ME)	Ivermectin	Yes	Yes

^1^ ME: microscopic examination, ^2^ AC: agar plate culture.

**Table 3 microorganisms-09-01905-t003:** Clinical features of patients who tested positive for *S. stercoralis* infection.

Clinical Signs/Symptoms	*n* (%)
Cough and wheezing	10 (18.2%)
Itch	10 (18.2%)
Nonspecific skin rash	7 (12.7%)
Abdominal pain	6 (10.9%)
Vomiting	3 (5.5%)
Urticaria	2 (3.6%)
Diarrhoea	2 (3.6%)
*Larva currens*	1 (1.8%)
Anorexia	1 (1.8%)

## Data Availability

The data presented in this study are available on request from the corresponding author. The data are not publicly available due to privacy restrictions.

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
