# Peer review of "Strongyloidiasis in Children Outside the Tropics: Do We Need to Increase Awareness?"

_microorganisms, 2021, doi:10.3390/microorganisms9091905_

Round 1
Reviewer 1 Report
This is a nice dataset describing the features of those children with positive strongyloides serology assessed in a single Italian centre. it highlights the prevalence of this disease, associations with eosinophilia, co-infections, general asymptomatic status, and response to treatment.
It needs to be reduced substantially to ensure clarity of message, and attention paid to definitions and to English.
Abstract: figures such as "75.5%) when used to describe small numbers, don't read well. Provide number, and % to nearest whole number
Intro line 42 - why "misdiagnosed"? do you mean "underdiagnosed"?
line 45: English
Intro too long: focus on strongy global epi; lack of data in children; large refugee influx to Italy/Europe in last 5 yrs; basoics of diagnostics and ease of treatment.
Poor specificity of serology not referenced - and arguable as there is no gold standard given low sensitivity of stool analysis. - suggest re-word.
line 72 ; drop "can be administered in multiple dose". just replace by "single dose is effective"
line 95: "dubious" wrong word - bordrline or equivocal better. In fact this needs to be defined clearly: you use the word borderline in methods but dubious in results.
Methods - apart from above - I think there should be more clarity on what this refugee clinic is and what the standard clinic infection screening algorithm is, that led to the positives.
Table 1 - not very interesting - suggest drop this.
line 192 - justification for positive serology being due to transplacental IgG not clear - it is expained below. Can this be reversed/not duplicated?
Toxocariasis. This is also a nematode and is often a false positive serological finding in context of a patient with another infection. So many of these may be false positive due to strongy. I think should be reworded as "positive toxocara serology" and this explanation put in text.
Section 3.2. Way too much detail in text. Numbers too small to say useful things. Suggest narrow down to
- Concurrent positive serologies and concurrent positive other infections
- Prevalence of eosinophilia
- Loa loa screening issue
- Symptoms
Why no data on where in Italy autochthonous cases likely to have been exposed? It is a big country with diverse ecology.
It is unclear why all positive patients had an USS. Suggest drop this
There should be a separate section on treatment
The lack of statistical significance of eos drop at 12 months needs explanation. Smaller sample size?
Reviewer 2 Report
The authors present a study of strongyloidiasis incidence and response to treatment in a population of children in Italy, both native and foreign-born. Although the study focuses on strongyloidiasis, it also describes the epidemiology of other infectious and parasitic diseases in this population, and so it may be useful to other clinicians working with a pediatric immigrant population. The conclusions on native-born Italian children are limited in their general applicability to other geographic areas.
In the definitions or discussion sections, the authors need to quantitatively indicate the poor sensitivity of a single stool sample to identify a case of Strongyloidiasis. A single stool exam is diagnostic in only one-third of patients. Serial stool examinations increase the sensitivity of stool exams but may be impractical. It is estimated that seven stool exams would provide close to 100% sensitivity (Siddiqui, A.A.; Genta, R.M.; Maguilnik, I.; Berk, S.L. Strongyloidiasis. In Tropical Infectious Diseases: Principles, Pathogens, and Practice, 3rd ed.; Guerrant, R.L., Walker, D.H., Weller, P.F., Eds.; Churchill Livingstone: Philadelphia, PA, USA, 2010; pp. 805–812; Nielsen, P.B.; Mojon, M. Improved diagnosis of Strongyloides stercoralis by seven consecutive stool specimens. Zentralbl. Bakteriol. Mikrobiol. Hyg. A 1987, 263, 616–618).
Considering the poor sensitivity and specificity of the tests used in the paper, perhaps “Probably resolved” should be “Possibly resolved.”
There are multiple corrections on English usage and spelling which need to be addressed, as indicated below.
investigators Line 45. The authors use the term: “Univocal” -which means “a term that has only one meaning.“ I think the authors intended to use the term: “Unequivocal”-meaning “leaving no doubt; unambiguous.”
Better: Ivermectin can be administered in multiple doses regimen even if a recent review highlights its high safety profile even below this weight [25]
line 76: Better: Albendazole, which has been considered an alternative drug for treatment of strongyloidiasis, has demonstrated unsatisfactory efficacy compared to ivermectin in several clinical trials [26-29].
Line 93: For each patient, data about age, gender and country of birth were collected.
Line 94: Better: Children were then divided in three groups depending on S. stercoralis serologic results: negative, dubious ,or positive (figure 1).
The authors use the term “dubious” to describe one of their serologic result groups, which is not an appropriate term. Dubious means “not to be relied upon; suspect.” A better term would be equivocal, meaning “uncertain.”
Line 99: For positive patients, clinical signs and symptoms, as well as past medical history, were evaluated.
Line 100: Better: Blood and faecal samples were obtained to perform serologies serologic assays for other NTDs (in accordance with patients’ risk factors) and parasitological examination or stool culture, to investigate the possible co-infection with other parasites.
Line 101: Better: Moreover, eosinophil cell counts and IgE levels, before and after treatment, were inscribed determined. Treatment information and response to target therapy were also assessed.
Line 106: Better: Strongyloidiasis was diagnosed in children with positive serologic results or stool test for S. stercoralis, regardless of eosinophil levels.
Line 108: serologic test, not serology
Line 112: Better: The criteria for “therapeutic failure” was if the worm was identified in stool samples or if the patient has a persistently positive serologic result 12 months after treatment. We defined as “probably cured” as children with negative coprological test and normalized eosinophil levels who did have follow-up serological test after been treated.
The criteria for ‘therapeutic failure” was the identification of the worm was identified in stool samples or a persistently positive serologic result 12 months after treatment.
Better: capitalize Laboratory: at the Microbiology and Virology Laboratory at Careggi University Hospital in Florence, Italy.
Line 130: Better: Data were analysed by SPSS Statistics 24.0 statistical software
Line 150: Better: Overall, 96.6% (2544/2633) children had negative serological results at the first evaluation.
Line 167: Peripheral levels of eosinophil; better: Peripheral eosinophil counts
Better: were within the normal range, except for three patients: two children who presented with eosinophilia (one infected by Schistosoma haematobium and another with elevated IgE), and one had moderate eosinophil counts but also had toxocariasis. 170 Considering those with two consequently dubious index, the only one receiving treatment was a girl born in Chile who had received 2 doses of ivermectin, though it wasn’t reported neither eosinophilia or any symptoms, with subsequent negative index more than12 months after treatment.
Dubious- not to be relied upon; suspect.
Equivocal- uncertain or questionable in nature.
Table 2; pirantel usually spelled pyrantel
Table 2, pruritis , instead of itch
Table 2, patient 4: hepatosplenic, not Epathosplenic
Table 3: itching or pruritis , instead of itch
Figure 3: hepatosplenic, instead of hepatic-splenic
Line 219, 222, 225: Space between S. and stercoralis
Line 228: itching instead of itch
Line 237: admitted instead of entrusted to an adult hospital
Line 248, space between 200 and mcg
Line 264: serologic
Line 264: Better: Regarding serologic negativization,
Line 266, 277, 282: Better: serologic test, NOT serology
Line 284: Better: …and immigrants, and no one none of the enrolled patients…
Line 287: Better: Herein, we have described a cohort of 2633 patients, mainly adopted or immigrated children, that were seen in a tertiary care children’s hospital in Italy, and were screened for strongyloidiasis by serologic testing.
Line 289: serologic test, NOT serology
Line 310: there is an error in this sentence: This could lead to an overestimation of prevalence in Italian children in our study foreign.???? Please reword.
Line 311: Hyphenate: foreign-born
Line 313: hyphenate human-to-human
Line 315: univocal- improper usage of this word; Better: but no specific data were available in their clinical records.
Line 326: Better: The higher rate of serologic positivity if compared to our study could be related to the higher chance of adults of being infected by S. stercoralis
Line 334: Serologic, NOT serology
Line 337: serologic, NOT serology
Line 338: Better: In particular, for ELISA tests, they suggested to set the limits to ≥2.5 and ≥2.2 for IVD and Bordier kit, respectively [40].
Line 341: space before [41].
351: serologic test or assay, not serology
353: pruritis, NOT hitch
356: Better: …underlying conditions, such as asthma, scabies….
- Better: In fact, it is well known that there is a high risk of polyparasitism in these patients
376: ivermectin does not need to be capitalized
379: Better: Recently, a trial on ivermectin scheme administration dosing in children
388: Better: A serologic assay was repeated
393: Better: with ivermectin due to a persistently positive titer
395: Better: treatment failure in cases of a persistently positive index
398: Better: the need of for a second treatment
399: better: … timing of follow-up and for the need for a second…
400: better: Our study has some limitations; first of all, it was retrospective. Due to this, the full clinical and hematochemical data were available only for children with positive serologic test. Therefore, it was not possible to establish which risk factors were associated with a higher rate of positivity, excepting for epidemiological data.
406: serologic test,
NOT serology
409 Serologic tests, NOT serology
414: Better: severe forms in the case of immunosuppression, it is important to identify and treat infected children
416: better: should be reserved to for high-risk groups,
422: better: Therefore, pediatricians should be aware about of this condition
423: better: S. stercoralis serologic testing was confirmed to be a useful tool, especially at first in the initial evaluation in patients coming from endemic areas. The A combination of more tests…
427: such as the management of treatment failure and long-term follow-up
